# Dynamic Analysis of a Micro Beam-Based Tactile Sensor Actuated by Fringing Electrostatic Fields

**DOI:** 10.3390/mi10050324

**Published:** 2019-05-14

**Authors:** Zhichong Wang, Qichang Zhang, Wei Wang, Jianxin Han

**Affiliations:** 1Department of Mechanics, School of Mechanical Engineering, Tianjin University, Tianjin 300072, China; wangzhichong@tju.edu.cn (Z.W.); wangweifrancis@tju.edu.cn (W.W.); 2Tianjin Key Laboratory of Nonlinear Dynamics and Control, Tianjin University, Tianjin 300072, China; hanjianxin@tju.edu.cn; 3Tianjin Key Laboratory of High Speed Cutting and Precision Machining, Tianjin University of Technology and Education, Tianjin 300222, China

**Keywords:** fringing electrostatic, nonlinear dynamic, tactile sensor

## Abstract

A new kind of fringing electrostatic actuation mode is developed. In this new actuation mode, the expression of fringing electrostatic force is found. The nonlinear dynamic analysis of this new actuation mode is presented by using the Method of Multiple Scales. An experiment is designed to observe the dynamic behaviors of this structure. It is observed that the resonance frequency rises with the increase of the initial displacement and the decrease of the slit gap; a smaller slit gap makes marked change of the resonance frequency in the same range of the initial displacement; the increase of the vibration amplitude is linear with the increase of the initial displacement; the fringing electrostatic force has a larger impact on the frequency response of the nonlinear vibration when the initial displacement, the beam length and the actuated voltage are larger. This new fringing electrostatic actuation mode can be used in a micro tactile sensor. The results of dynamic analysis can provide support for sensor design. Based on the dynamic investigations into the micro cantilevered beam actuated by fringing electrostatic force; three usage patterns of the sensor are introduced as follows. Firstly, measuring resonance frequency of the micro beam can derive the initial displacement. Second, the initial displacement can be derived from vibration amplitude measurement. Third, jump phenomenon can be used to locate the initial displacement demand.

## 1. Introduction

Tactile sensors, most commonly referred to as strain and pressure sensors, can collect mechanical property data from the human body and the local environment, and then provide valuable insights into the human health status or artificial intelligence systems [1,2,3]. Meanwhile, it can also be equipped on robots in order to be aware of their surroundings, keep away from potentially destructive effects and provide information for subsequent tasks such as in-hand manipulation [4,5]. Change of resistance, capacitance, electrical charge and optical distribution can be used in various sensing systems [6], and the typical sensing mechanisms for tactile sensors includes piezo-resistive, capacitive, piezo-electric and optical. Compared with other types of tactile sensors, capacitive sensors have high sensitivity and fast frequency response [7,8,9]. Thus, many resonance capacitive sensors, which are very sensitive to small changes, are designed and optimized [10,11,12,13]. 

The electrostatic actuation method is an important method of Micro electromechanical systems (MEMS). The system actuated by the electrostatic force is parametric excitation system [14,15,16]. Among the electrostatic actuation structures, parallel-plates configuration is widely used in MEMS [17,18,19,20,21,22]. However, this actuation method can lead to instabilities like pull-in [23,24,25]. In order to obtain large amplitude and avoid the instabilities, the fringing-field actuation technique has got its attention. This actuation approach has several advantages, including the ability to obtain large amplitude displacements without the limitation of the proximity of the electrodes and the possibility of significantly tuning the resonant frequency response range [26]. This structure has attracted attention and has been studied by some communities. Experimental and theoretical analysis of micro-cantilevers actuated by fringing-field electrostatics was performed [26]. The different behaviors of curved micro beam with low initial elevation and relatively high initial elevation were studied when actuated by fringing-field electrostatic force [27]. A parametrically excited electrostatic resonator, which had a flexible support, used weaker electrostatic fringe fields to get higher vibrational amplitude [28]. These researches show that the device actuated by the fringing electrostatic force can reveal many new static and dynamic behaviors. 

In the present work, researchers always care about the situations when the beam thickness is near the electrode thickness [26]. In these cases, the relationship between the electrostatic force and the initial displacement is linear when the beam vibrates near the middle of the electrode in the thickness direction. However, we find that the relationship between the electrostatic force and the initial displacement is nonlinear when the thickness of the electrode is much more than the thickness of the beam. This situation can reveal some new dynamic behaviors. Based on the dynamic investigations into the micro cantilevered beam actuated by fringing electrostatic force, the working principle and the usage patterns of a new micro tactile sensor are presented in this paper. 

The paper is organized as follows. In Section 2, a new resonance tactile sensor is designed and the dynamic modeling of a micro cantilevered beam actuated by fringing electrostatic fields is given. This section consists of three parts: in Section 2.1, the working principle of the micro tactile sensor is presented; in Section 2.2, the fringing electrostatic force and the influences of geometric parameters on this force are found; in Section 2.3, the governing equation of this new kind of fringing electrostatic actuation mode is outlined. In Section 3, the dynamic characteristics of the micro cantilevered beam which actuated by fringing electrostatic force are analyzed. In Section 4, an experiment is designed to observe the dynamic analysis of this structure. In Section 5, the dynamic behaviors of this structure are summarized and the usage patterns of this micro tactile sensor are presented. 

## 2. Problem Formulation and Dynamic Modeling 

### 2.1. Working Principle

The proposed micro tactile sensor consists of a micro cantilevered beam and a pair of micro electrodes. The concept structure of the micro tactile sensor is shown in Figure 1. One end of the micro cantilevered beam is fixed on the base. The edge of the micro electrode is also connected to the base. The base is nonconductive. The micro cantilevered beam and the micro electrode are conductive, but they are disconnected. When the electrostatic voltage is applied to the micro beam and the micro electrode is connected to the ground wire, the fringing electrostatic force appears between the micro beam and the micro electrode. Under the actuation of the electrostatic force, the micro beam vibrates. When the pressure *P* is applied to the micro electrode, the parts which link electrode and base deform linearly, the micro electrode moves from position 1 to position 2. Then the relative position between the micro cantilevered beam and the micro electrode changes $d_{p}$. This relative position change leads to the change of fringing electrostatic force, which effects the vibration behaviors of the micro beam. In this way the vibration behaviors of the micro beam can reflect the pressure applied to the micro electrode. Some obvious advantages are as following: a large deflection can be obtained without the limitation of the proximity of the electrodes; this structure which is compatible with the circuit can be designed to smaller scale; a smaller scale leads to a better sensitivity. 

It is the key issue to grasp the relative position change effects the vibration behaviors of the micro cantilevered beam. The paper focuses on the influence of the relative position change in the vibration behaviors of the micro cantilevered beam. This can provide support for sensor design. Figure 2 is the schematic illustration of micro cantilevered beam and micro electrode. In this picture, $l_{b}$
$w_{b}$ and $t_{b}$ denote the length, width and thickness of the micro beam respectively; $l_{s}$
$w_{s}$ and $t_{s}$ denote the length, width and thickness of the micro electrode respectively; $d_{g}$ denotes the slit gap in the width direction; $d_{}$ denotes the initial displacement in the thickness direction, it is the placement position of the beam. In order to study the vibration behaviors of the micro beam, the geometric parameters of the structure are taken as $l_{b}=l_{s}=5\;\mathrm{mm}$, $w_{b}=0.4\;\mathrm{mm}$, $t_{b}=0.01\;\mathrm{mm}$, $w_{s}=1.5\;\mathrm{mm}$, $t_{s}=0.3\;\mathrm{mm}$ and $d_{g}=0.04\;\mathrm{mm}$. The range of the initial displacement is less than half of the electrode thickness. 

### 2.2. Fringing Electrostatic Force 

In this paper, the thickness of the electrode is much more than the thickness of the beam, as shown in Figure 2. The distributed electrostatic force is $q_{e}=f_{e}\times v_{e}^{2}$, in which $f_{e}$ is electrostatic force on the beam per unit length per square voltage; $v_{e}\left(t\right)$ is a combined DC/AC voltage applied on the beam, i.e., $v_{e}\left(t\right)=v_{\mathrm{DC}}+v_{\mathrm{AC}}\cos\omega_{e}t$. 

When the cantilevered beam is in a position of the fringing electrostatic actuation structure, the voltage applied to the cantilevered beam is defined as $+v_{e}$, the voltage applied to the electrode is defined as $-v_{e}$, and the capacitance between the cantilevered beam and electrode is $4C_{e}$. Then, the charge on the cantilevered beam is $+8C_{e}v_{e}$, The charge on the electrode is $-8C_{e}v_{e}$. Assume that the charges on the cantilevered beam are evenly distributed at the two points, and the charges on the electrode are evenly distributed at the eight points, as shown in Figure 2. Then, the charge of each point on the cantilevered beam is $q_{b}=+4C_{e}v_{e}$, and the charge of each point on the electrode is $q_{s}=-C_{e}v_{e}$. According to Coulomb’s law and the geometric position of the plate, the electrostatic force acting on the cantilevered beam in the direction of thickness is
(1)$$F_{e}=2k_{e}q_{b}q_{s}\left(\begin{array}{l} r_{11}{}^{-2}\sin\theta_{11}+r_{21}{}^{-2}\sin\theta_{21}+r_{12}{}^{-2}\sin\theta_{12}+r_{22}{}^{-2}\sin\theta_{22}+\\ r_{13}{}^{-2}\sin\theta_{13}+r_{23}{}^{-2}\sin\theta_{23}+r_{14}{}^{-2}\sin\theta_{14}+r_{24}{}^{-2}\sin\theta_{24} \end{array}\right)$$

See Appendix A for the symbolic meaning. 

In the expression of electrostatic force—the proportion of $r_{12}{}^{-2}\sin\theta_{12}$ and $r_{22}{}^{-2}\sin\theta_{22}$ are the largest. When the fringing electrostatic force near *d* = 0 is concerned, the Taylor expansion of the above approximate expression near *d* = 0 is
(2)$$F_{e}=2k_{e}q_{b}q_{s}\left\{\left[\frac{C_{1}}{C_{2}}+\frac{1}{3}\frac{t_{s}\left(6t_{s}/C_{2}{}^{5/2}-1.75t_{s}C_{1}/C_{2}\right)}{C_{2}}\right]d^{3}+\left[\frac{2}{C_{2}{}^{3/2}}-\frac{1.5t_{s}{}^{2}}{C_{2}{}^{5/2}}\right]d\right\}$$
in which $C_{1}=3.75t_{s}{}^{2}/C_{2}{}^{5/2}-3/C_{2}{}^{3/2}$, $C_{2}=d_{g}{}^{2}+0.25t_{s}{}^{2}$.

From this Taylor expansion, we can see that the expression of fringing electrostatic force near *d* = 0 can be fitted by polynomials containing the first and third terms of the distance *d*. 

By using the finite element software, Figure 3 shows the equipotential lines when actuated voltage of 1 Volt is applied across the two electrodes and the beam for d=0 mm, d=0.02 mm, d=0.04 mm, d=0.06 mm, respectively. When d=0 mm, the equipotential lines are symmetrical around the beam, When the initial displacement increases, the equipotential lines become unsymmetrical, which leads to the appearance of the fringing electrostatic force. 

Under different electrode thickness, the values of fringing electrostatic force responding to initial displacement are calculated. In Figure 4, the data points and fitted curves of fringing electrostatic force per unit length are recorded. When the thickness of the electrode is much greater than the thickness of the beam, the electrostatic force increases monotonically and nonlinearly. Since the thickness of the electrode dominates the deflection range of the beam, the thickness of the electrode should not be too small. Thus, the fit function of the fringing electrostatic force responding to initial displacement should consist of linear term and nonlinear term in this case. The polynomial fit function which includes linear and cubic parameters is proposed, and the fringing electrostatic force is approximated as $f_{e}=r_{1}d+r_{3}d^{3}$, in which $r_{1}$ and $r_{3}$ are fitting parameters. The assumption of the fringing electrostatic force can reflect the trend of electrostatic force change, and this assumption is convenient for analysis. The beam vibrates around the initial displacement *d*, the fringing electrostatic force responding to the vibration amplitude u can be written as $f_{e}=r_{1}\left(d+u\right)+r_{3}{\left(d+u\right)}^{3}=e_{p}\left(1+e_{p1}u+e_{p2}u^{2}+e_{p3}u^{3}\right)$, in which $e_{p}=r_{1}d+r_{3}d^{3}$, $e_{p1}=\left(r_{1}+3r_{3}d^{2}\right)/\;e_{p}$, $e_{p2}=3r_{3}d/\;e_{p}$, $e_{p3}=r_{3}/\;e_{p}$. 

The fit function of electrostatic force in this paper is different from that in reference [26]. Figure 5 shows the fitted curves of electrostatic force based on different fit function when $t_{s}=0.3\;\mathrm{mm}$. In this figure, FC. 0 is the fitted curves which based on this paper, FC. 1 and FC. 2 are the fitted curves which based on reference [26]. As shown in Figure 5, FC. 1 and FC. 2 cannot fit the data when d = 0 mm – 0.30 mm. The beam vibrates in the range of the thickness of the electrode, i.e., d = 0 mm – 0.15 mm, which is what we care about in this paper. FC. 0 can fit the data when d = 0 mm – 0.15 mm. It means that the fit functions, which based on reference [26], cannot fit the electrostatic force in this case. The fit function in this paper can reflect the change of the electrostatic force, when the vibration amplitude of the cantilevered beam is less than the thickness of the electrode. 

### 2.3. Dynamic Modeling

The influences of the initial displacement change in the beam’s dynamic behaviors are focused in this paper. By using the elastic beam theory, the equation of motion and the boundary conditions of the beam are written as [26]
(3)$$EI\frac{\partial^{4}u(x,t)}{\partial x^{4}}+\rho w_{b}t_{b}\frac{\partial^{2}u(x,t)}{\partial t^{2}}=q_{e}+q_{a}$$
(4)$$u\left(0,t\right)=\frac{\partial u(0,t)}{\partial x}=0,\;\frac{\partial^{2}u(l_{b},t)}{\partial x^{2}}=\frac{\partial^{3}u(l_{b},t)}{\partial x^{3}}=0$$
where u(x,t) is the deflection of the beam in the thickness direction; *E* is the Young’s modulus; $I=w_{b}t_{b}^{3}/12$ is the inertial moment of the beam cross section; ρ is the mass density of the beam. The material parameters of the beam, which is made of brass, are taken as E=108 GPa and $\rho =8500\;\mathrm{kg}/m^{3}$. Actuation force $q_{e}$ and damping force $q_{a}$ represent the fringing electrostatic force and the aerodynamic force per unit length respectively. The distributed aerodynamic force $q_{a}$ is [29,30] $\left|q_{a}\right|=0.5\rho_{a}w_{b}c_{a}{\left(\partial u/\partial t\right)}^{2}$, in which the direction of the aerodynamic force is opposite to the direction of the velocity; $c_{a}$ is the drag coefficient; $\rho_{a}$ is the density of the air. 

For analytical convenience, we obtain the following non-dimensional equation of motion and the boundary conditions.
(5)$$\frac{\partial^{4}U}{\partial X^{4}}+\frac{\partial^{2}U}{\partial T^{2}}={Ε}_{0}\left(1+{Ε}_{1}U+{Ε}_{2}U^{2}+{Ε}_{3}U^{3}\right){\left(1+V_{AC}\cos W_{e}T\right)}^{2}-Α\frac{\partial U}{\partial T}\left|\frac{\partial U}{\partial T}\right|$$
(6)$$U\left(0,T\right)=\frac{\partial U(0,T)}{\partial X}=0,\frac{\partial^{2}U(1,T)}{\partial X^{2}}=\frac{\partial^{3}U(1,T)}{\partial X^{3}}=0$$

See Appendix B for the symbolic meaning. 

We use the Galerkin discretization method to transform the partial differential equation to the ordinary differential equations [31]. The model in this paper is based on the fundamental frequency vibration. The steady-state solution of the non-dimensional governing equation is written by U(X,T)=Φ(X)Θ(T), and the mode function Φ(X) for cantilevered beam is $\Phi \left(X\right)=\mathrm{ch}\lambda_{r}X-cos\lambda_{r}X+\xi_{r}\left(\mathrm{sh}\lambda_{r}X-\sin\lambda_{r}X\right)$, in which $\lambda_{r}=1.875$, $\xi_{r}=-\left(\mathrm{ch}\lambda_{r}+\cos\lambda_{r}\right)/\left(\mathrm{sh}\lambda_{r}+\sin\lambda_{r}\right)$. Substituting U(X,T)=Φ(X)Θ(T) into the governing equation. Then multiplying the outcome by the mode function, and integrating the resultant equation from X = 0 to 1. An ordinary differential equation with respect to time is obtained as
(7)$$\begin{array}{l} \frac{\partial^{2}\Theta }{\partial T^{2}}+\alpha_{k}\Theta =\\ \alpha_{Ep}\left(1+\alpha_{ep1}\Theta +\alpha_{ep2}\Theta^{2}+\alpha_{ep3}\Theta^{3}\right)\left[\left(1+\frac{V_{AC}{}^{2}}{2}\right)+2V_{AC}\cos W_{e}T+\frac{V_{AC}{}^{2}}{2}\cos2W_{e}T\right]-\alpha_{a}\frac{\partial \Theta }{\partial T}\left|\frac{\partial \Theta }{\partial T}\right| \end{array}$$

See Appendix C for the symbolic meaning. 

The deflection splits into a static deflection and a dynamic deflection, i.e., $\Theta \left(T\right)=\Theta_{0p}+\vartheta_{p}\left(T\right)$, then we obtain static equation and dynamic equation.
(8)$$S_{0}+S_{1}\Theta_{0p}+S_{2}\Theta_{0p}^{2}+S_{3}\Theta_{0p}^{3}=0$$
(9)$$\vartheta_{p}^{\prime \prime }+K_{1}\vartheta_{p}+K_{2}\vartheta_{p}{}^{2}+K_{3}\vartheta_{p}{}^{3}+\alpha_{a}\vartheta_{p}^{\prime }\left|\vartheta_{p}^{\prime }\right|=\left[\begin{array}{l} \left(F_{E1}\cos W_{e}T+F_{E2}\cos2W_{e}T\right)+\\ \left(K_{E1P1}\vartheta_{p}+K_{E1P2}\vartheta_{p}{}^{2}+K_{E1P3}\vartheta_{p}{}^{3}\right)\cos W_{e}T+\\ \left(K_{E2P1}\vartheta_{p}+K_{E2P2}\vartheta_{p}{}^{2}+K_{E2P3}\vartheta_{p}{}^{3}\right)\cos2W_{e}T \end{array}\right]$$
where (′) denotes the derivate with respect to T. See Appendix D for the symbolic meaning. 

## 3. Dynamic Analysis 

The static equation is a one variable cubic equation, which can be solved by using the method of reference [32]. The nonlinear dynamic equation is solved by using the Method of Multiple Scales (MMS). Then the dynamic equation can be written by
(10)$$\vartheta_{p}^{\prime \prime }+K_{1}\vartheta_{p}+\varepsilon K_{2}\vartheta_{p}{}^{2}+\varepsilon K_{3}\vartheta_{p}{}^{3}+\varepsilon \alpha_{a}\vartheta_{p}^{\prime }\left|\vartheta_{p}^{\prime }\right|=\varepsilon \left[\begin{array}{l} \left(F_{E1}\cos W_{e}T+F_{E2}\cos2W_{e}T\right)+\\ \left(K_{E1P1}\vartheta_{p}+K_{E1P2}\vartheta_{p}{}^{2}+K_{E1P3}\vartheta_{p}{}^{3}\right)\cos W_{e}T+\\ \left(K_{E2P1}\vartheta_{p}+K_{E2P2}\vartheta_{p}{}^{2}+K_{E2P3}\vartheta_{p}{}^{3}\right)\cos2W_{e}T \end{array}\right]$$
where ε is regarded as a small non-dimensional bookkeeping parameter, σ is a detuning parameter, $W_{e}{}^{2}=K_{1}+\varepsilon \sigma$. 

Introduce $W_{0}^{2}=\alpha_{k}$, when the static deflection is small and σ=0, the resonance frequency ratio is
(11)$$W_{e}{}^{2}/W_{0}{}^{2}=K_{1}/\alpha_{k}={Κ}_{w0}-{Κ}_{w2}d^{2}$$
in which the coefficients are
(12)$${Κ}_{w0}=1-r_{1}\frac{{\left(\int_{0}^{1}\Phi dX\right)}^{2}}{\int_{0}^{1}\frac{\partial^{4}\Phi }{\partial X^{4}}\Phi dX}\frac{l_{b}{}^{4}v_{DC}^{2}}{EI}\left[1+\frac{1}{2}{\left(\frac{v_{AC}}{v_{DC}}\right)}^{2}\right],\;{Κ}_{w2}=3r_{3}\frac{{\left(\int_{0}^{1}\Phi dX\right)}^{2}}{\int_{0}^{1}\frac{\partial^{4}\Phi }{\partial X^{4}}\Phi dX}\frac{l_{b}{}^{4}v_{DC}^{2}}{EI}\left[1+\frac{1}{2}{\left(\frac{v_{AC}}{v_{DC}}\right)}^{2}\right]$$

The primary resonance of nonlinear dynamic equation is analyzed, yields
(13)$$\begin{array}{l} -W_{e}\alpha_{0}\beta_{0}{}^{\prime }=\frac{1}{2}\sigma \alpha_{0}-\frac{1}{4}K_{3}\alpha_{0}{}^{3}+\\ \left(\frac{1}{2}F_{E1}+\frac{1}{4}K_{E1P2}\alpha_{0}{}^{2}\right)\cos\beta_{0}+\left(\frac{1}{4}K_{E2P1}\alpha_{0}+\frac{1}{4}K_{E2P3}\alpha_{0}{}^{3}\right)\cos2\beta_{0} \end{array}$$
(14)$$-W_{e}\alpha_{0}{}^{\prime }=-\frac{W_{e}^{2}}{\pi }\alpha_{a}\alpha_{0}{}^{2}+\frac{1}{2}F_{E1}\sin\beta_{0}+\left(\frac{1}{4}K_{E2P1}\alpha_{0}+\frac{1}{8}K_{E2P3}\alpha_{0}{}^{3}\right)\sin2\beta_{0}$$
where $\alpha_{0}$ is the amplitude of $\vartheta_{p}$, $\beta_{0}$ is the phase difference with $W_{e}$, (′) denotes the derivate with respect to $T_{1}=\varepsilon T$. The complete proof is given in Appendix E. 

The steady-state periodic motion corresponds to the solution of the system of equations, by conditions $\alpha_{0}^{\prime }=0$ and $\beta_{0}^{\prime }=0$. Finally, the frequency response equation of the primary resonance can be derived as
(15)$${\left[\frac{-P_{C1}-\sqrt{P_{C1}{}^{2}-8P_{C2}\left(P_{C0}-P_{C2}\right)}}{4P_{C2}}\right]}^{2}+{\left[\frac{-4P_{C2}P_{S0}}{4P_{C2}P_{S1}-2P_{C1}P_{S2}-2P_{S2}\sqrt{P_{C1}{}^{2}-8P_{C2}\left(P_{C0}-P_{C2}\right)}}\right]}^{2}=1$$
in which the coefficients are$P_{C0}=\frac{1}{2}\sigma \alpha_{0}-\frac{1}{4}K_{3}\alpha_{0}{}^{3}$, $P_{C1}=\frac{1}{2}F_{E1}+\frac{1}{4}K_{E1P2}\alpha_{0}{}^{2}$, $P_{C2}=\frac{1}{4}K_{E2P1}\alpha_{0}+\frac{1}{4}K_{E2P3}\alpha_{0}{}^{3}$, $P_{S0}=-\frac{W_{e}^{2}}{\pi }\alpha_{a}\alpha_{0}{}^{2}$, $P_{S1}=\frac{1}{2}F_{E1}$, $P_{S2}=\frac{1}{4}K_{E2P1}\alpha_{0}+\frac{1}{8}K_{E2P3}\alpha_{0}{}^{3}$.

The relationship between the electrostatic force and the initial displacement is nonlinear. It can reveal nonlinear dynamic behaviors of the vibration beam. When $v_{\mathrm{DC}}=30\;V$ and $v_{\mathrm{AC}}=0.5\;V$, based on the frequency response equation, the frequency response curve of the primary resonance is shown in Figure 6. It exhibits linear behavior. Based on Equation (11) and setting σ = 0, i.e., when the vibration beam resonates, the relationship between the square of resonance frequency and that of initial displacement is linear, as shown in Figure 7. From Figure 7, the resonance frequency rises, with the increase of the initial displacement and the decrease of the slit gap. In the same range of the initial displacement, a smaller slit gap makes marked change of the resonance frequency. So, the initial displacement can be obtained from resonance frequency measurement. What’s more, decreasing the slit gap can enhance the sensitivity of the sensor to the initial displacement. 

The vibration peak value can be found by the frequency response equation, Figure 8 shows the effects of initial displacement and slit gap on the vibration amplitude of the primary resonance. Increase of the vibration amplitude is linear, with the increase of the initial displacement. Linearity is an important parameter for sensors because linearity indicates a directly proportional relationship between output and input signals of a sensing system [1]. So, it is available that the initial displacement is obtained from vibration amplitude measurement of the primary resonance. 

When $v_{\mathrm{DC}}=300\;V$ and $v_{\mathrm{AC}}=5\;V$, the nonlinear dynamic equation Equation (9) is solved by using the MMS and the Fourth-Order Runge–Kutta Method (RK4) [33], and the effect of the initial displacement on the vibration amplitude is shown in Figure 9. The results of the MMS and the RK4 are in a good agreement when the amplitude is small. The error between the results of the MMS and the RK4 increases, when the vibration amplitude increases. Figure 9 proves the results of the MMS verified. As shown in Figure 9, when *d* = 0.03 mm, there are two values of the vibration amplitude when *W_e_ / W_0_* = 1.0750 and 1.0775. The jump phenomenon has been found, when the initial displacement increases, the jump frequency increases.

Primary resonance’s vibration amplitude versus initial displacement under different excitation frequency ratio is shown in Figure 10. In Figure 10, the frequency ratio is 1.0750, 1.0775 and 1.0800 respectively. Within the range of initial displacement [0.01 mm, 0.05 mm], the amplitude is calculated once every 0.002 mm interval, and the bifurcation diagram of amplitude with respect to initial displacement is drawn. It can be clearly observed that the number of equilibrium points of parameter *u* changes with the change of parameter *d*. For the case of frequency ratio 1.0750, parameter *u* has three equilibrium points when parameter *d* = 0.03; for the case of frequency ratio 1.0775, parameter *u* has three equilibrium points when parameter *d* = 0.032 and 0.034; for the case of frequency ratio 1.0800, parameter *u* has three equilibrium points when parameter *d* = 0.034 and 0.036. With the change of parameter *d*, there is a jump phenomenon in parameter *u*. When the jump phenomenon occurs, the corresponding excitation frequency is the resonance frequency of the vibration system. 

By comparing the frequencies of the three equilibrium points, it is found that with the increase of initial displacement, the smaller excitation frequencies first appear three equilibrium points. Because there is a corresponding relationship between the excitation frequency ratio and the initial displacement of the cantilever beam when there are three equilibrium points in the vibration system. Therefore, the initial displacement of the cantilever beam can be described by measuring the excitation frequency of the three equilibrium points. So, jump phenomenon can be used to locate the demand initial displacement. The excitation frequency can be changed to adjust the change of the demand initial displacement. 

When *d* = 0.03 mm and $W_{e}/W_{0}$ = 1.075, phase trajectory is drawn in Figure 11. It can be seen from the figure that the amplitude of the cantilever beam is related to its initial state, that is, when the initial state energy is high, the amplitude of the cantilever beam corresponds to the higher equilibrium point S1; when the initial state energy is low, the amplitude of the cantilever beam corresponds to the lower equilibrium point S3. The equilibrium point S2 is unstable saddle. 

The jumping amplitude change can be detected more easily and more quickly. In the following, the jump phenomenon is studied, based on the frequency response equation. And the impact of the different parameters, which include the material parameters, the beam length and the actuated voltage, on the nonlinear dynamic characteristic is presented. 

The effects of the initial displacement and the material parameters on the frequency response curve of the primary resonance are shown in Figure 12. The frequency ratio $W_{e}/W_{0}$ of the vibration peak value of the aluminum beam, whose Young’s modulus is the least, is the biggest. That because that when the Young’s modulus decreases, the value of the parameter $E_{0}$ increases, which leads the increase of the frequency ratio $W_{e}/W_{0}$. It means that the fringing electrostatic force has a largest impact on the resonance frequency of the aluminum beam. As show in this figure, the amplitude u of the brass beam, whose density is the largest, is the biggest. The nonlinear response of the brass beam is obvious. That because that when the density increases, the value of the parameter $\alpha_{a}$ decreases, which leads the decrease of the damping and the increase of the vibration amplitude. To increase the vibration amplitude, we use the brass beam. 

When $l_{s}=6\;\mathrm{mm}$, the effects of the initial displacement on the frequency response curve of the primary resonance are shown in Figure 13. Compared with Figure 9, the vibration amplitude u and frequency ratio $W_{e}/W_{0}$ of the vibration peak value are larger, the nonlinear behavior is more obvious. Meanwhile, with the increase of the initial displacement, the increase of the jump frequency changes is more obviously. It means that the fringing electrostatic force has a larger impact on the frequency response, when the length of beam is larger.

When $v_{\mathrm{DC}}=350\;V$
$v_{\mathrm{AC}}=5\;V$ and $v_{\mathrm{DC}}=300\;V$
$v_{\mathrm{AC}}=10\;V$, Figure 14 shows the frequency response curve of the primary resonance under different initial displacement. Compared with Figure 9, when DC/AC voltage are big enough, the electrostatic force can lead to obvious nonlinear vibration. Furthermore, as the increase of the actuated voltage, the nonlinear vibration strengthens. When a larger amplitude is expected, the actuated voltage is always set big enough, and the nonlinear vibration must be considered. 

## 4. Experimental Setup and Results 

An experiment is designed to observe the dynamic analysis of this structure. The experimental setup for the dynamic tests consists of excitation powers (high voltage power and waveform generation), mechanical parts (cantilevered beam and electrode) and detection parts (laser displacement sensor and oscilloscope). The schematic of experimental setup is depicted in Figure 15. Because the experimental conditions are limited, the geometric parameters of the structure are magnified, which are taken as $l_{b}=50\;\mathrm{mm}$, $w_{b}=4\;\mathrm{mm}$, $t_{b}=0.1\;\mathrm{mm}$, $l_{s}=50\;\mathrm{mm}$, $w_{s}=15\;\mathrm{mm}$, $t_{s}=3\;\mathrm{mm}$, $d_{g}=0.5\;\mathrm{mm}$ and $v_{\mathrm{DC}}=300\;V$, $v_{\mathrm{AC}}=5\;V$. 

The cantilevered beam is directly actuated by a periodic wave produced from a waveform generator, while DC voltage produced by high voltage power is applied to the cantilevered beam at the same time. The electrode is connected with ground. In this way, both DC voltage and AC voltage are applied between the cantilevered beam and electrode. The moving platform can change the initial displacement between the cantilevered beam and electrode. 

The vibration of the cantilevered beam is more obvious, when the excitation frequency is close to the resonance frequency. And the vibration amplitude is the largest, when the excitation frequency equals to the resonance frequency. The laser displacement sensor is used to detect the vibration amplitude, by transforming the amplitude change to voltage change. Then this voltage change signal is given to the oscilloscope. The oscilloscope is used to display and record the data of both frequency and amplitude. It means that the resonance frequency change can be detected by using both the laser displacement sensor and the oscilloscope. 

When $v_{\mathrm{DC}}=300\;V$ and $v_{\mathrm{AC}}=5\;V$, Figure 16 shows that the resonance frequency rises with the increase in initial displacement, in which *d* = 0 mm – 0.8 mm. As the same as the theory analysis, it is expected that relationship between the input (the square of initial displacement) and the output (the square of resonance frequency) is linear, when the beam is far from the middle and the end of the electrode in the thickness direction. In this range, the initial displacement can be obtained from resonance frequency measurement. 

The vibration amplitude of the cantilevered beam is relatively small, when the cantilevered beam is near the middle of the electrode in the thickness direction. It is not easy to find the resonance frequency. So, the error is large near this point in Figure 16. 

When the beam is near the end of the electrode in the thickness direction, the frequency response curves are shown in Figure 17. When $v_{\mathrm{DC}}=100\;V\;\mathrm{and}\;200\;V$, the frequency response curve is linear., while the amplitude of the beam is relatively small. But when $v_{\mathrm{DC}}=300\;V$, the frequency response curve is nonlinear, while the nonlinearity of the system is of the softening type, as shown in [26]. If the amplitude of the beam is large enough, the electrostatic force does not agree with linear and cubic fit function at all, when the beam is near the end of the electrode in the thickness direction. So, it causes an error near the end of the electrode in Figure 16. 

When the excitation frequency equals to the resonance frequency, the vibration amplitude is the largest. The largest vibration amplitude values responding to different initial displacement are recorded by the test. When the initial displacement is 0.4 mm to 0.8 mm, the vibration amplitudes which are given by the MMS and the TEST are shown in Figure 18. When the initial displacement is 0.4 mm to 0.8 mm, the increase in the vibration amplitude is linear, and the initial displacement can be obtained from amplitude measurement. As show in Figure 18, the result of the MMS agrees well with the experimental result. Base on Figure 8, the measurement error of slit gap leads the error between the results of the MMS and the TEST in Figure 18. 

In the above test, the hardening effect is not obvious, since the cubic fitting parameter is relatively small. Figure 13 shows that the fringing electrostatic force has a larger impact on the frequency response, when the length of beam is larger. To increase cubic fitting parameter, the slit gap is reduced to 0.3 mm, and another beam with $l_{b}=100\;\mathrm{mm}$ is used. The frequency response curve exhibits hardening behavior, and there is a jump in 5.140 Hz, as shown in Figure 19. 

## 5. Conclusions 

In this paper, a new fringing electrostatic actuation mode is developed. Some obvious advantages of this new actuation mode are as following: a large deflection can be obtained without the limitation of the proximity of the electrodes; this structure which is compatible with the circuit can be designed to smaller scale; a smaller scale leads to better sensitivity. Through the combination of theoretical modeling, analytical calculation, numerical verification and experimental research, the mechanism of fringing electrostatic force and the complex response law of fringing electrostatic actuation vibration system are revealed, the relationship between system parameters and vibration response is analyzed, the application scheme of this actuation mode is put forward.

The fringing electrostatic force and the dynamic investigations into the micro cantilevered beam actuated by fringing electrostatic force are of great concern. Through analysis, the expression of fringing electrostatic force is found; the effects of the some parameters on the dynamic behaviors are investigated. Results shows that the fringing electrostatic force is nonlinear, which leads to nonlinear vibration of the micro-cantilevered beam; the resonance frequency rises with the increase of the initial displacement and the decrease of the slit gap; in the same range of the initial displacement, a smaller slit gap makes marked change of the resonance frequency; with the increase of the initial displacement, the increase of the vibration amplitude is linear; when the initial displacement increases, the jump frequency increases; the fringing electrostatic force has a larger impact on the frequency response, when the length of beam is larger; as the increase of the actuated voltage, the nonlinear vibration strengthens. 

Moreover, that are the influences of initial displacement change in the dynamic behaviors of the micro cantilevered beam that helps us to design a new micro tactile sensor. This sensor can measure the pressure based on the initial displacement change. The initial displacement can be derived by measuring resonance frequency and vibration amplitude of the micro cantilevered beam. And the jump phenomenon can be used to locate the initial displacement. 

## Figures and Tables

**Figure 1 micromachines-10-00324-f001:**
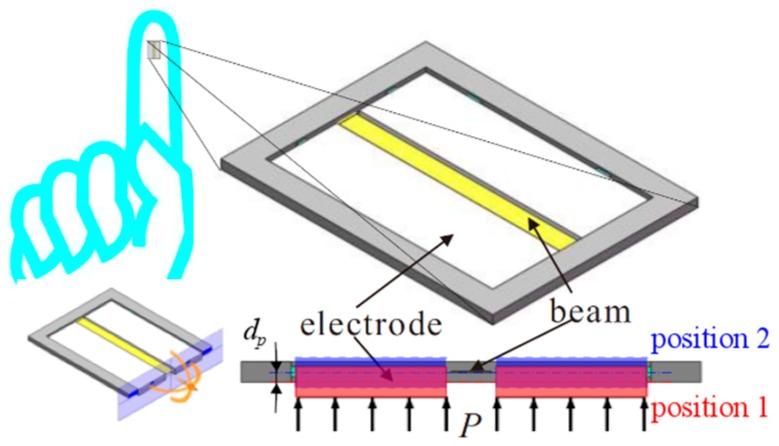
Concept structure of the tactile sensor.

**Figure 2 micromachines-10-00324-f002:**
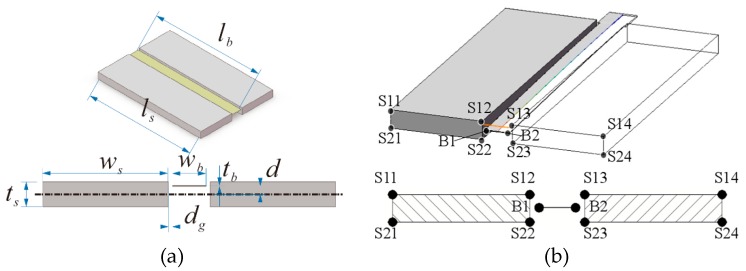
Schematic illustration of cantilevered beam and electrode. (**a**) Structure size. (**b**) Simplified model.

**Figure 3 micromachines-10-00324-f003:**
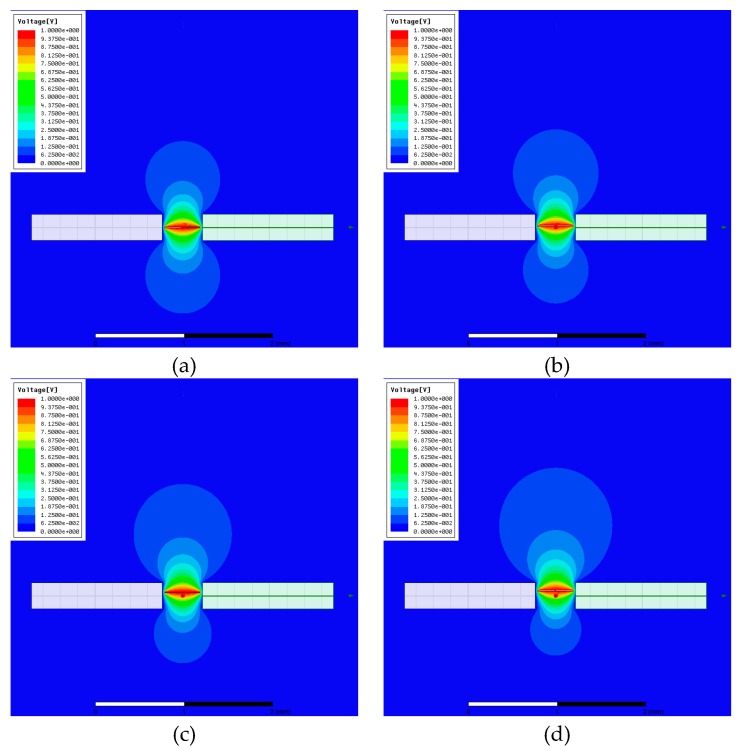
Equipotential lines around the beam and the electrodes assuming $v_{e}=1\;V$ and (**a**) for d=0 mm, (**b**) for d=0.02 mm, (**c**) for d=0.04 mm, (**d**) for d=0.06 mm.

**Figure 4 micromachines-10-00324-f004:**
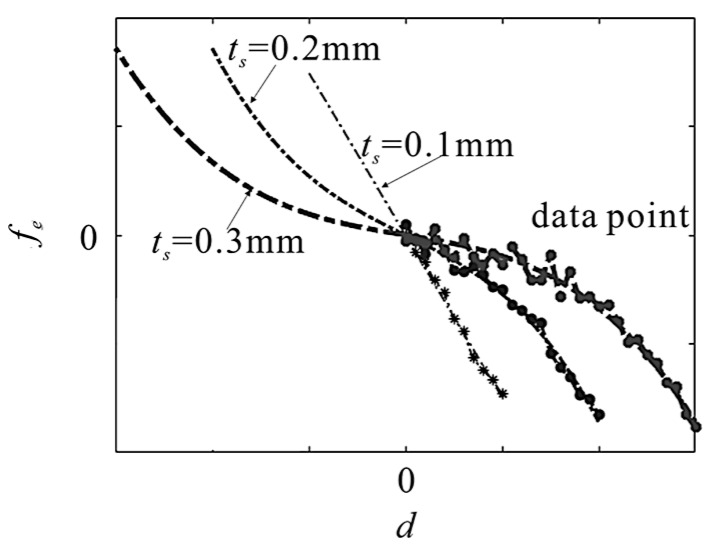
Data points and fitted curves of electrostatic force responding to initial displacement under different electrode thickness.

**Figure 5 micromachines-10-00324-f005:**
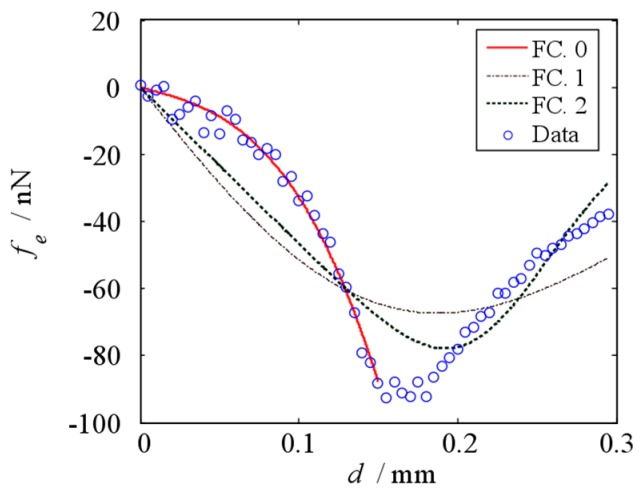
Fitted curves of electrostatic force based on different fit function.

**Figure 6 micromachines-10-00324-f006:**
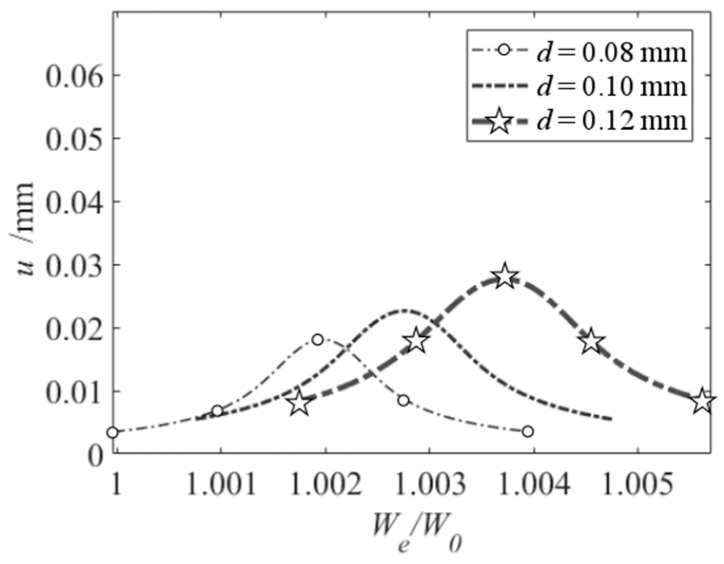
Primary resonance’s frequency response curve under different initial displacement when $v_{\mathrm{DC}}=30\;V$ and $v_{\mathrm{AC}}=0.5\;V$.

**Figure 7 micromachines-10-00324-f007:**
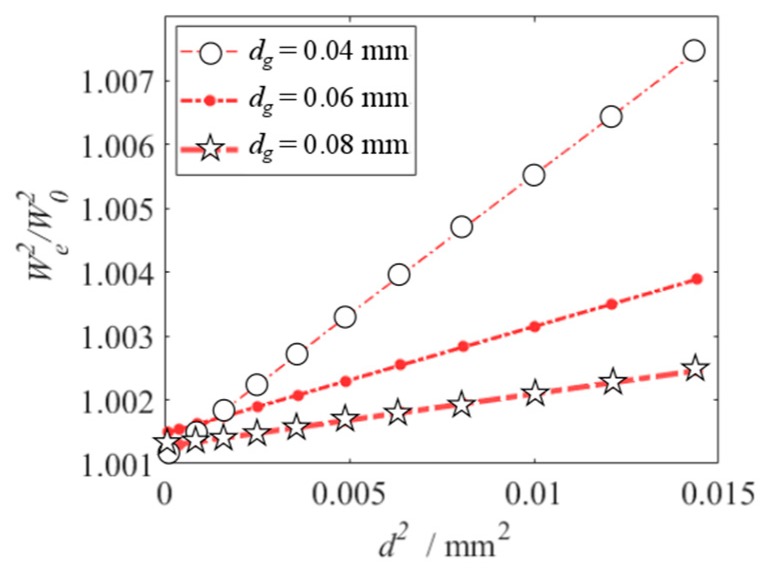
Primary resonance’s resonance frequency versus initial displacement under different slit gap.

**Figure 8 micromachines-10-00324-f008:**
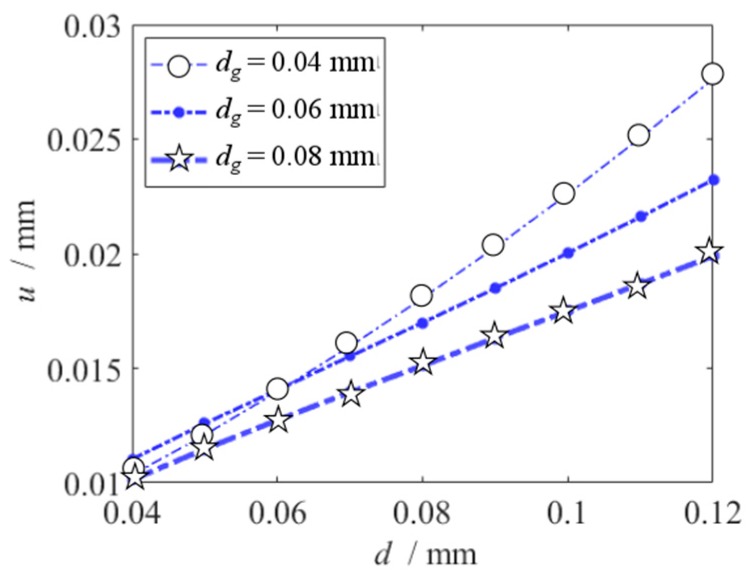
Primary resonance’s vibration amplitude versus initial displacement under different slit gap.

**Figure 9 micromachines-10-00324-f009:**
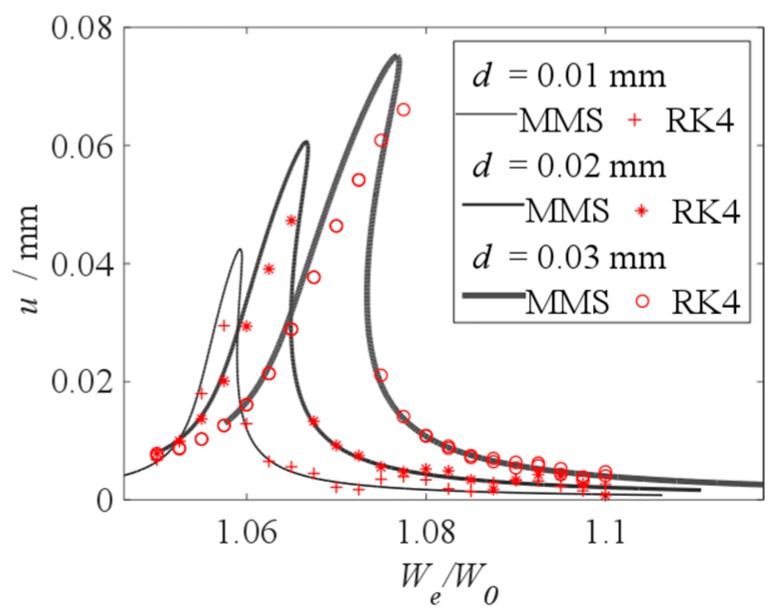
Vibration amplitude which is solved by using Method of Multiple Scales (MMS) and RK4 when $v_{\mathrm{DC}}=300\;V$ and $v_{\mathrm{AC}}=5\;V$.

**Figure 10 micromachines-10-00324-f010:**
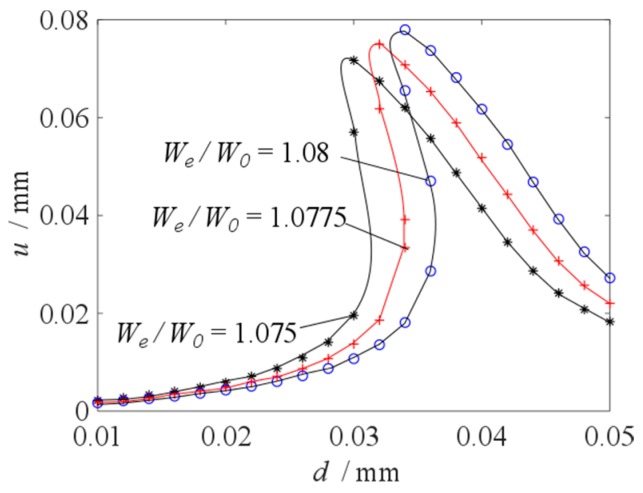
Primary resonance’s vibration amplitude versus initial displacement under different excitation frequency ratio.

**Figure 11 micromachines-10-00324-f011:**
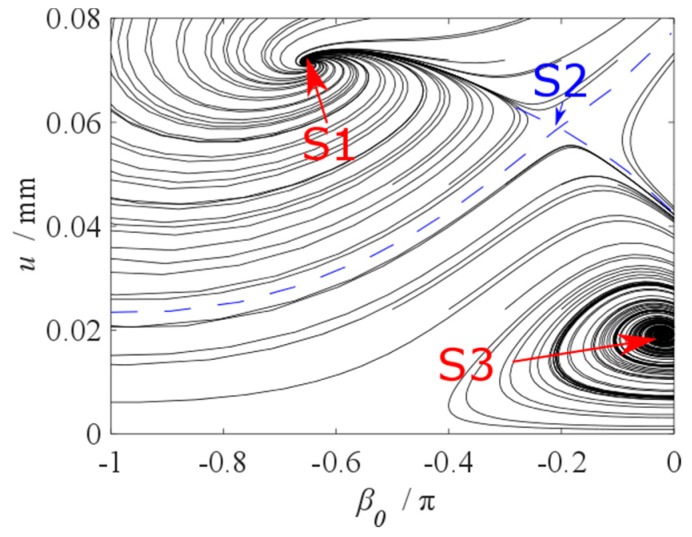
Phase trajectory.

**Figure 12 micromachines-10-00324-f012:**
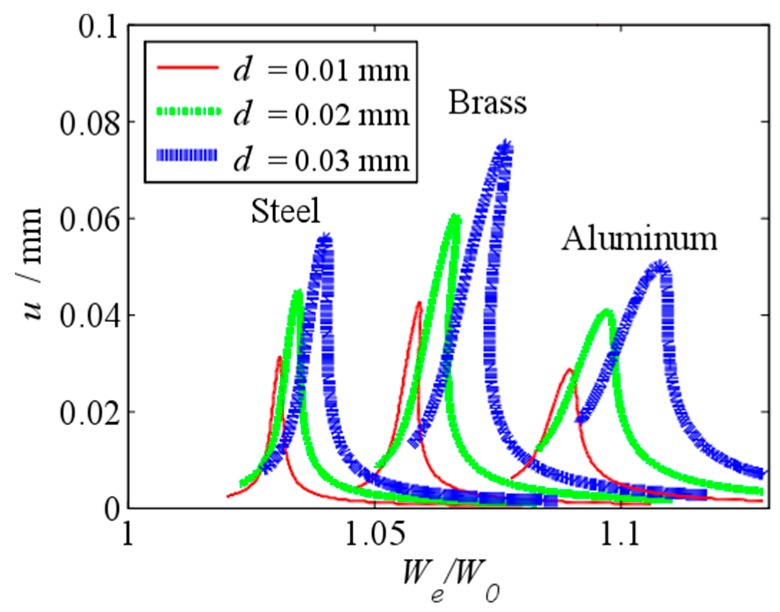
Primary resonance’s frequency response curve under different initial displacement when the beam is made of steel, brass and aluminum.

**Figure 13 micromachines-10-00324-f013:**
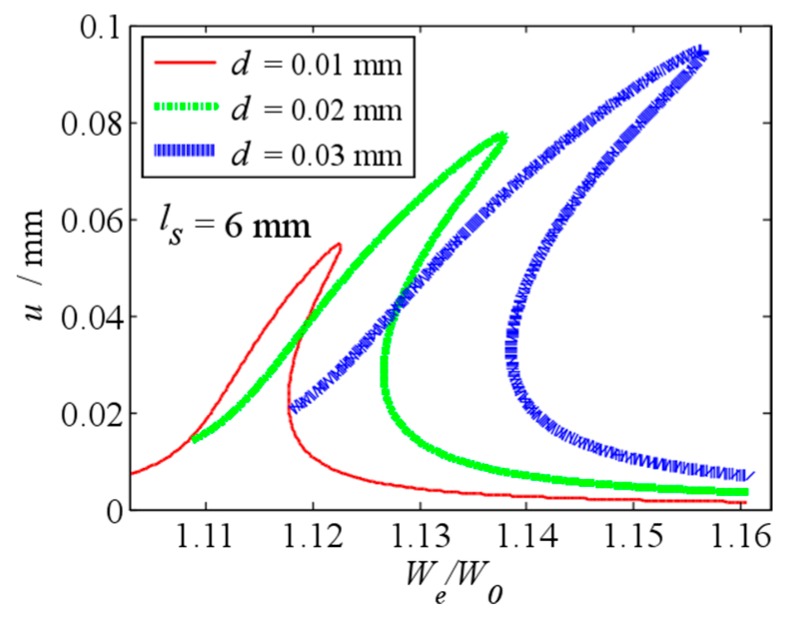
Primary resonance’s frequency response curve under different initial displacement when $l_{s}=6\;\mathrm{mm}$.

**Figure 14 micromachines-10-00324-f014:**
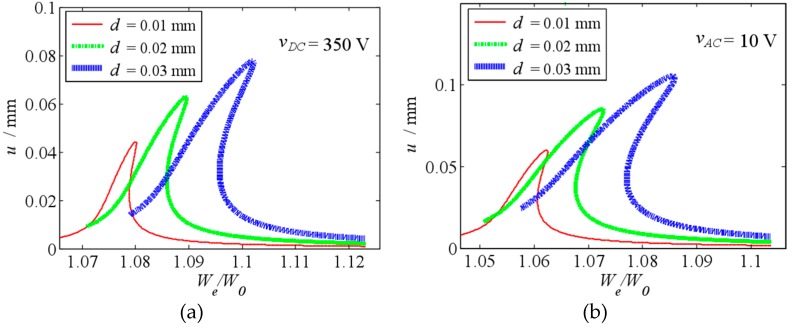
Primary resonance’s frequency response curve under different initial displacement (**a**) when $v_{\mathrm{DC}}=350\;V$ and $v_{\mathrm{AC}}=5\;V$ (**b**) when $v_{\mathrm{DC}}=300\;V$ and $v_{\mathrm{AC}}=10\;V$.

**Figure 15 micromachines-10-00324-f015:**
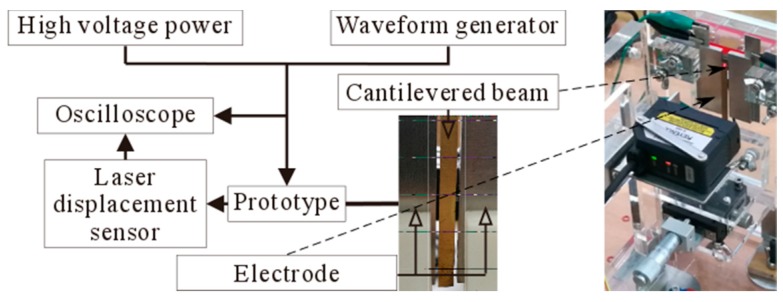
Experimental setup.

**Figure 16 micromachines-10-00324-f016:**
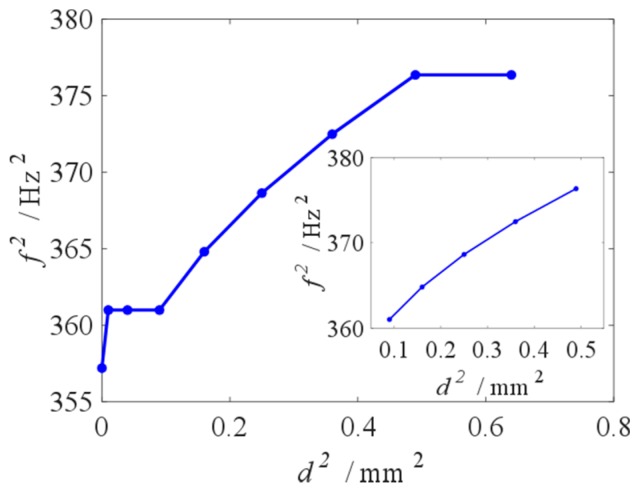
Resonance frequency versus initial displacement (linear region magnification).

**Figure 17 micromachines-10-00324-f017:**
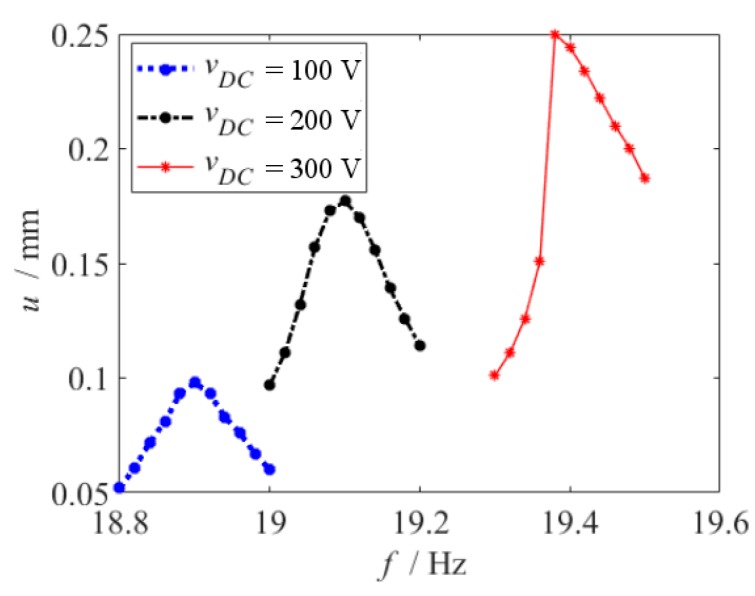
Frequency response curve when the beam is near the end of the electrode in the thickness direction.

**Figure 18 micromachines-10-00324-f018:**
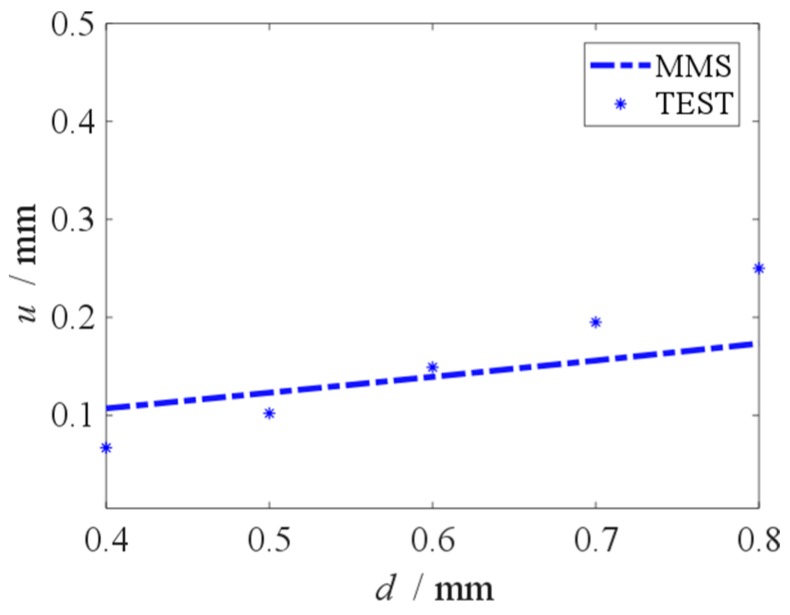
Vibration amplitude versus initial displacement base on MMS and TEST.

**Figure 19 micromachines-10-00324-f019:**
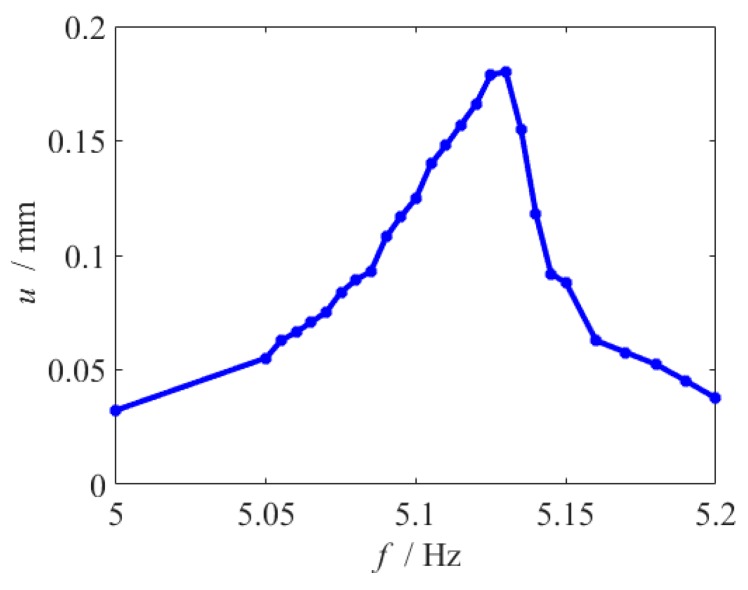
Frequency response curve when $l_{b}=100\;\mathrm{mm}$ and $d_{g}=0.3\;\mathrm{mm}$.

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
