# Peer review of "Dynamic Analysis of a Micro Beam-Based Tactile Sensor Actuated by Fringing Electrostatic Fields"

_micromachines, 2019, doi:10.3390/mi10050324_

Round 1

Reviewer 1 Report

The paper present a study of the vibrations of a cantilever micro beam employed in a tactile sensor. The system is based on the fringing electrostatic force obtained by means of a combined DC/AC voltage. A finger pressure modifies the position of the micro beam and so the vibrations changes.

The paper use a variety of methods to study the complex system, spanning from Finite Element, to theoretical modeling and experimental tests. However, there are some important concerns regarding the work.

Line 132. The polynomial fit function includes linear and cubic parameters to simulate the fringing electrostatic force. Is there a theoretical justification for that? What about the forces expressed at lines 136 and 137?

The simulated material is brass, steel and Aluminum. Why the Authors are choosing such materials? Brass and steel for example are not used in microfabrication or MEMS technology.

The aerodynamic force depends on the square of the velocity and so the differential equation is not linear. The Authors stated this circumstance in the text, but still the mode function Phi(x) does not depend on the physical aerodynamic parameters. It is also not clear what is the meaning of steady state solution: is it the same equation with null time derivatives? In this case how to deal with the nonlinear aerodynamic force? Alternatively, maybe the analysis concerns a transient state? This is not clear.

The results obtained by means of the theoretical approach are not well related to the original purpose of the paper, which is the development of the tactile sensor. Please, explain these relations.

The numerical results simulate an applied voltage up to 300 V. This value seems very great for the microsystem. Is there a proof that the air gap may sustain such a difference in potential?

The experimental tests are performed on a one order of magnitude scaled system (it seems a x 10 factor). Of course the results will be useless unless the scaling effects are considered and quantified.

Finally, the conclusion is very confusing and the reader misses grasping the point of the whole investigation. It must be rewritten.

Minor comments:

Improve English language and Figures

Reference [36] is not complete.

Line 164. Please reformulate the sentence to avoid starting the sentence with qe and qa

Author Response

Dear Reviewer,

We thank you for your careful reading and thoughtful comments on previous draft. We have carefully adopted your comments in our revision. Thanks for your help, this paper is clearer and more compelling.

Best wishes,

Yours sincerely

Qichang Zhang

Reviewer 2 Report

Dear Editor.

I clearly and honestly could not see why this work is needed. What does in innovate with respect to the known literature in the past 6-25 years? Everything reported here is well-known about the electrostatic fringing field actuation of micro and meso cantilevers. We know that the actuation is nonlinear, and nothing new is anticipated and/or demonstrated experimentally other than dependencies already known and trivial on all parameters of the system, such as the gap, size of the electrode compared with beam thickness, etc. I could not find a single innovative thing in the manuscript.

I would encourage the authors to bring to the table something new about electrostatic actuation, at least somewhere in the manuscript, and emphasize sharply and exactly what is novel, or at least lacking in current depth of the existing literature within the field – in abstract, introduction, and summary; currently nothing here, in my opinion, should warrant publication.

Formally, I am not sure that references 28, 29, 32, are relevant here, as they are applicable for bridge rather than caltilever beams. In a future revision I propose the authors to consider replacing these references with other, more relevant, ones relating to the cantilever geometries that they study.

Author Response

(The authors gave the same response as above.)

Round 2

Reviewer 1 Report

English Style must be improved.

Furthermore, section 4 must contain some comments about the test dimensional scaling  

Reviewer 2 Report

Dear Editor.

Thanks to the authors for addressing most of my concerns, as the abstract and conclusions were indeed rephrased to stress the apparent innovation and emphasize how they combine the three principles of actuation and detection modes within the same analysis and experiments.

The authors decided to leave references 28-29, 32. I still think that they are redundant here. Also, relating to the parallel-plate configuration referred to in line 44-45, it seems to me again that reference 14 is redundant, whereas the other references 18-23.

In summary it is my opinion that references 28-29,32,14 should be removed or replaced with more appropriate references within the scope and framework of the current research.